# Eyelid-Aware Region-of-Interest Constraint for Meibomian Gland Segmentation

**Dawoon Lee**[1]                                          DAW00N@SCH.AC.KR
**Seula Kye**[2]                                           KYESEULA@SCH.AC.KR
**Onseok Lee**[*1,2]                                       LEEOS@SCH.AC.KR

[1] *Dept. of Medical IT Engineering, Soonchunhyang University, Asan, Republic of Korea.*

[2] *Dept. of Software Convergence, Graduate School, Soonchunhyang University, Asan, Republic of Korea.*

## Abstract

Automatic segmentation of meibomian glands in infrared meibography is challenging due to low contrast, specular reflections, and variability in eyelid positioning. Existing studies typically perform gland segmentation directly on the full image; However, without considering anatomical constraints, these predictions are highly susceptible to background noise and irrelevant structures, which limits their clinical reliability. In this work, we propose an eyelid-aware segmentation pipeline that explicitly utilizes anatomical eyelid information. The proposed method first segments the eyelid to define a region of interest(ROI), and then performs meibomian gland segmentation strictly within this region. Experimental results using multiple U-Net-based models under identical conditions show that while Dice and IoU scores are comparable across models, whereas significant differences are observed in boundary-based metric, with ResU-Net++ achieving the lowest boundary error. These findings highlight the importance of anatomical constraints and boundary-aware evaluation.

**Keywords:** Meibomian Gland Segmentation, Infrared Meibography, Deep Learning.

## 1. Introduction

Meibomian Gland Dysfunction (MGD) is a major cause of dry eye disease, and structural changes in meibomian glands(MGs) are important indicators for diagnosis and severity assessment (Saha et al., 2022). Infrared meibography enables non-invasive visualization of gland morphology.

However, MGs are thin and elongated structures with low contrast and are highly affected by specular reflections, making automatic segmentation difficult. In addition, variations in eyelid positioning lead to inconsistent gland representation, further degrading segmentation reliability.

Conventional approaches perform gland segmentation directly on the full image, which makes them susceptible to interference from regions outside the eyelid, such as skin folds, eyelashes, and reflections (Saha et al., 2022).

To address these limitations, we propose an eyelid-aware pipeline that leverages eyelid information. As illustrated in Figure 1, the eyelid is first segmented to define a valid Region of Interest(ROI), and gland segmentation is then performed within this region to reduce background interference. In addition, boundary-aware evaluation is incorporated to better assess the segmentation of thin structures (Karimi and Salcudean, 2020).

---

* Contributed equally

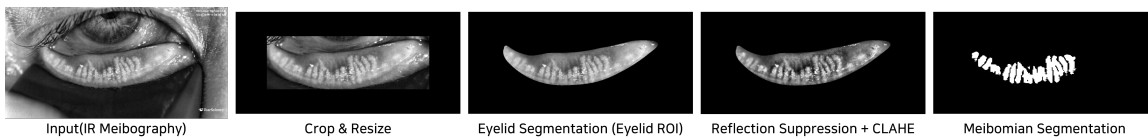

| Input(IR Meibography) | Crop & Resize | Eyelid Segmentation (Eyelid ROI) | Reflection Suppression + CLAHE | Meibomian Segmentation |

Figure 1: Overview of the proposed eyelid-aware pipeline.

## 2. Methods

### 2.1. Preprocessing

Each infrared meibography image is cropped to remove irrelevant background regions. Padding and resizing are then applied to standardize the input dimensions across all samples.

### 2.2. Eyelid Segmentation and ROI Extraction

Eyelid segmentation is performed using a U-Net architecture with a ResNet-18 encoder(Ronneberger et al., 2015)(Park et al., 2026). The input size is 528×1168×3, and the network predicts two classes: eyelid and background. The model is trained using the Adam optimizer for 60 epochs with a mini-batch size of 8.

The predicted eyelid mask defines the ROI (Figure 1), suppressing non-relevant areas such as specular reflections, skin texture, and eyelashes, thereby enforcing anatomically valid constraints prior to gland segmentation.

### 2.3. Meibomian Gland Segmentation

MG segmentation is performed within the eyelid ROI. Reflection suppression and contrast enhancement using Contrast Limited Adaptive Histogram Equalization (CLAHE) are applied to improve gland visibility while preserving structural details. Multiple U-Net-based architectures are trained and evaluated under identical data split conditions.

Segmentation performance is evaluated using Dice coefficient(Dice), Intersection over Union(IoU), and 95th percentile Hausdorff distance(HD95). Dice and IoU measure overlap accuracy, while HD95 captures boundary discrepancies, a feature that is critical for thin structures(Karimi and Salcudean, 2020). Experiments are conducted on the MGD-1K dataset.

## 3. Results

### 3.1. Eyelid Segmentation Results

The eyelid segmentation model achieves high accuracy and stable boundary prediction, enabling reliable extraction of anatomically valid ROIs for subsequent processing(Table 1).

Table 1: Eyelid Segmentation Performance of U-Net–Based Models

| Global Accuracy | Mean IoU | BFScore |
|---|---|---|
| 0.98161 | 0.95637 | 0.92649 |

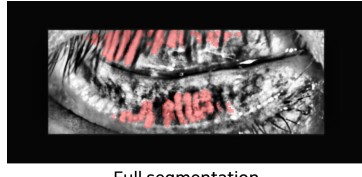 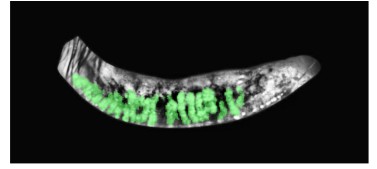 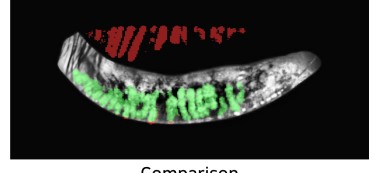

Full segmentation      Pipeline segmentation      Comparison

Figure 2: Comparison between full-image segmentation and the proposed pipeline.

### 3.2. Meibomian Gland Segmentation Results

All evaluated models show comparable performance in Dice and IoU. However, clear differences are observed in HD95 (Table 2), indicating variation in boundary precision. ResU-Net++ achieves the lowest HD95, demonstrating the most accurate and stable boundary delineation. This indicates that overlap-based metrics alone are insufficient to fully characterize segmentation performance.

Qualitative improvements are also observed in Figure 2. The proposed method produces more continuous gland structures with reduced boundary discontinuities and artifacts.

As illustrated in Figure 2, the proposed eyelid-aware pipeline suppresses predictions in anatomically irrelevant regions and constrains segmentation within the eyelid. This leads to improved robustness, particularly in cases with strong specular reflections and low contrast, resulting in more consistent and reliable segmentation.

Table 2: Segmentation Performance of U-Net–Based Models

| Model | Evaluation Metrics | | |
|---|---|---|---|
| | Dice | IoU | HD95 |
| U-Net | 0.8260 | 0.7050 | 5.08px |
| ResU-Net++ | 0.8363 | 0.7199 | 4.83px |
| Attention U-net | 0.8281 | 0.7086 | 5.62px |

## 4. Conclusion

We proposed an eyelid-aware pipeline that incorporates anatomical constraints for meibomian gland segmentation. The proposed method improves segmentation robustness by restricting predictions to anatomically valid regions.

Experimental results highlight that boundary-aware metrics (HD95) reveal differences not captured by overlap-based measures, emphasizing the importance of boundary precision.

This work provides a foundation for future quantitative analysis and clinical applications, including automated meibomian gland dropout assessment.

## Acknowledgments

This research was supported by the MSIT(Ministry of Science, ICT), Korea, under the National Program for Excellence in SW, supervised by the IITP(Institute of Information & communications Technology Planning & Evaluation) in 2026(2021-0-01399).

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
