# OpenReview forum: "Eyelid-Aware Region-of-Interest Constraint for Meibomian Gland Segmentation"
_MIDL.io/2026/Short_Papers — MIDL 2026 - Short Papers Poster_

### Official Review · Reviewer_NKiZ · 2026-05-06
**Simple and practical idea with clinical relevance, but limited novelty and depth of evaluation**

**Rating:** 2
**Confidence:** 5

**Review:**

- This is a relevant problem in ophthalmic imaging, particularly for MGD assessment where gland morphology is important but difficult to capture due to low contrast and imaging artifacts. The idea of constraining segmentation using an anatomically meaningful ROI (eyelid) is intuitive and practically useful.
- From a methodological standpoint, the approach is straightforward: a two-stage pipeline where eyelid segmentation defines the ROI, followed by gland segmentation. This is a reasonable design and aligns well with clinical structure. However, the overall contribution is relatively incremental. ROI-based filtering is a common strategy, and the paper does not clearly position how this differs from prior work beyond applying it in this specific context.
- One positive aspect is the consistent experimental setup across multiple U-Net variants, which helps isolate the effect of architecture on performance. The observation that Dice/IoU are similar while HD95 differs is useful and reinforces the argument that boundary metrics are important for thin structures.
- That said, I feel evaluation is somewhat limited. The paper does not include a clear comparison between full-image segmentation vs. ROI-constrained segmentation in quantitative terms (only qualitative mention). This is a key missing piece since the main claim is about the benefit of ROI constraints. Also, improvements are relatively modest, and it is not clear how statistically significant they are.
- Another limitation is the lack of ablation or deeper analysis. For example, how sensitive is the pipeline to errors in eyelid segmentation? Does ROI masking ever remove valid gland regions? These aspects are important for practical deployment but are not discussed.

**Summary:**

This paper proposes an eyelid-aware segmentation pipeline for meibomian gland (MG) segmentation in infrared meibography. The method first segments the eyelid to define an anatomically valid region of interest (ROI), and then performs gland segmentation within this constrained region. The approach is evaluated using multiple U-Net-based architectures under the same setup, showing similar Dice/IoU performance but noticeable differences in boundary quality measured by HD95. Results suggest that enforcing anatomical constraints improves robustness and that boundary-based metrics are more informative for thin structure segmentation. The work emphasizes the importance of ROI design and evaluation metrics in this domain.

**Strengths:**

- Addresses a clinically relevant and challenging problem (MG segmentation in low-quality imaging conditions).
- Simple and intuitive pipeline that leverages anatomical structure (eyelid) effectively.
- Clear motivation for using boundary-aware metrics like HD95 for thin structures.
- Consistent evaluation across multiple architectures under the same conditions.
- Qualitative results suggest improved robustness and reduced artifacts.
- Easy to implement and potentially useful in real clinical pipelines.

**Weaknesses:**

- Limited novelty; ROI-based constraint is a well-known idea and not deeply extended here.
- Missing key quantitative comparison between full-image vs. ROI-based segmentation.
- No ablation studies (e.g., effect of ROI quality, preprocessing steps like CLAHE).
- Limited analysis of failure cases or robustness (e.g., incorrect eyelid segmentation).
- Improvements are relatively small and lack statistical significance analysis.
- Method section lacks depth and detailed justification of design choices.
- Evaluation is restricted to a single dataset, limiting generalizability claims.

**Justification Of Rating:**

The paper presents a practical and clinically meaningful idea with a clean implementation, and the emphasis on boundary-aware evaluation is a good contribution. However, the novelty is limited and the experimental validation does not fully support the central claims, particularly the benefit of the ROI constraint. With stronger quantitative comparisons, ablations, and deeper analysis, this could be a solid short paper. In its current form, it sits at borderline.

---

### Decision · Program_Chairs · 2026-05-08

Accept (Poster)